# Overcoming Resource Constraints in Federated Learning: Large Models Can Be Trained with only Weak Clients

**Yue Niu, Saurav Prakash** *  
*Department of Electrical and Computer Engineering*  
*University of Southern California*

*yueniu,sauravpr@usc.edu*

**Souvik Kundu**  
*Intel Labs, San Diego, USA*

*souvikk.kundu@intel.com*

**Sunwoo Lee**  
*Department of Computer Science and Engineering,*  
*Inha University*

*sunwool@inha.ac.kr*

**Salman Avestimehr**  
*Department of Electrical and Computer Engineering*  
*University of Southern California*

*avestime@usc.edu*

**Reviewed on OpenReview:** *https://openreview.net/forum?id=lx1WnkL9fk*

## Abstract

Federated Learning (FL) is emerging as a popular, promising decentralized learning framework that enables collaborative training among clients, with no need to share private data between them or to a centralized server. However, considering many edge clients do not have sufficient computing, memory, or communication capabilities, federated learning of large models still faces significant bottlenecks. To keep such *weak but crucial* clients in the loop, prior works either consider a heterogeneous-client setting where clients train models with different sizes; or offload training to the server. However, the heterogeneous-client setting requires some clients to train full model, which is not aligned with the resource-constrained setting; while the latter ones break privacy promises in FL when sharing intermediate representations or labels with the server. To overcome these limitations, in this work, we formulate a realistic, but much less explored, cross-device FL setting in which no client can train a full large model nor is willing to share any intermediate information with the remote server. Under such a formulation, we develop a principal sub-model (PriSM) training methodology to collaboratively train a full large model, while assigning each client a small sub-model that is a *probabilistic* low-rank approximation to the full server model. When creating sub-models, PriSM first performs a principal kernel analysis in the orthogonal kernel space to obtain *importance* of each kernel. Then, PriSM adopts a novel importance-aware sampling process to select a subset of kernels (i.e., a kernel with high importance is assigned with a higher sampling probability). This sampling process ensures each sub-model is still a *low-rank approximation* to the full model, while all sub-models together achieve *nearly full coverage* on the principal kernels. To further improve memory efficiency while still preserving accuracy, PriSM also exploits low-rank structure in inter-mediate representations and allows each sub-model to learn only a subset of them. Our evaluations on various datasets and models (CNNs, LSTMs, Transformers) under different resource-constrained settings demonstrate that PriSM yields an accuracy improvement of up to 10% compared to existing works. More importantly, PriSM does not incur significant accuracy degradation compared to full-model training (e.g., only $\sim 2\%$ accuracy drops for ResNet-18/CIFAR-10 when clients train only $0.2\times$ sub-models). Code is available at https://github.com/yuehniu/modeldecomp-fl.

---

*Authors have equal contribution.

# 1 Introduction

Federated Learning (FL) is emerging as a popular paradigm for decentralized and privacy-preserving machine learning as it allows clients to perform ML optimization jointly without directly sharing local data (McMahan et al., 2017; Kairouz et al., 2021). Thus, it enables privacy protection on local data, and leverages distributed local training to attain a better global model. This creates opportunities for many edge devices rich in data to participate in joint training without data sharing. For example, resource-limited smart home devices can train local vision or language models using private data, and achieve a server model that generalizes well to all users via FL (Pichai, 2019).

Despite significant progress in FL in the recent past, several crucial challenges still remain when moving to the edge. In particular, limited computation, memory, and communication capacities prevent clients from learning large models for leveraging vast amounts of local data at the clients. This problem is getting increasing attention in current literature (Diao et al., 2021; Horvath et al., 2021; Yao et al., 2021; Hong et al., 2022; Mei et al., 2022; Vepakomma et al., 2018; He et al., 2020). For example, recent works propose a sub-model training methodology by assigning clients with different subsets of the full model depending on their available resources (Diao et al., 2021; Horvath et al., 2021; Yao et al., 2021; Li et al., 2021). However, these works have an underlying assumption that some of the clients have sufficient resources to train a nearly full large model. In particular, methods like FedHM (Yao et al., 2021) that adapt low-rank compression to FL incur more memory footprint for intermediate representations, even for small sub-models compared to the full model training. As a result, server model size is limited by the clients with maximum computation, memory, and communication capacities. To overcome resource constraints on clients, other works (Vepakomma et al., 2018; Thapa et al., 2022; He et al., 2020) change the training paradigm by splitting a model onto server and clients. The computational burden on the clients is therefore relieved as the dominant part of the burden is offloaded to the server. However, such a methodology requires sharing of intermediate representations and/or labels with the server, which directly leaks input information and potentially compromises privacy promises of FL (Erdoğan et al., 2022; Zhang et al., 2020).

**Our Contributions**. Unlike prior works, this work considers even more constrained and realistic settings at the edge, in which no client is capable of training a large model nor is willing to share any intermediate data and/or labels with the server. To this end, we propose a novel P̲rincipal S̲ub-M̲odel (PriSM) training methodology. At a high level, PriSM *allows each client to only train a small sub-model, while still attaining a full model on the server and achieving comparable accuracy as the standard training.*

The cornerstone of PriSM is the models' inherent low-rank structure, which is commonly used in reducing compute costs (Khodak et al., 2021; Denton et al., 2014). However, naive low-rank approximation in FL (Yao et al., 2021), where all clients only train top-$k$ kernels, incurs a notable accuracy drop, especially in very constrained settings. In Figure 1, we delve into the matter by showing the number of principal kernels required in the orthogonal space to accurately approximate each convolution layer in the first two *ResBlocks* in ResNet-18 (He et al., 2016) during FL training[1]. We observe that even at the end of the FL training, around half of the principal kernels are still needed to sufficiently approximate each convolution layer. We have similar findings for the remaining convolution layers (See Sec 4.4). Therefore, to avoid the reduction in server model capacity, it is essential to ensure that all server-side principal kernels are collaboratively trained on clients, especially when each client can only train a very small sub-model (e.g., $< 50\%$ of the server model).

Based on the above observations, we develop an effective probabilistic strategy to select a subset of kernels and create a sub-

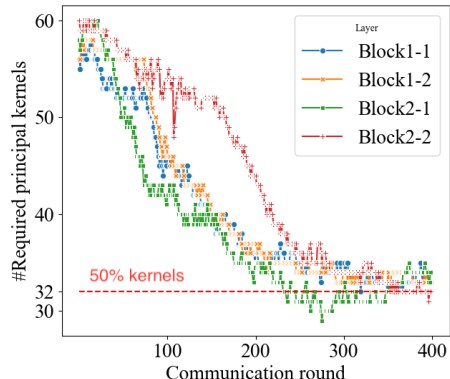

Figure 1: #principal kernels in the orthogonal space required to accurately approximate convolution layers in the first two ResBlocks in ResNet-18 during FL training. Block*i-j* indicates *j*-th convolution layer in *i*-th ResBlock. The server model gradually attains a low-rank structure. However, even in the final model, half of the principal kernels are still required to approximate most layers.

---

[1]See Sec 4.4 for further details, especially for calculating the required number of principal kernels.

model for each client as shown in Figure 2. More specifically, PriSM first converts the model into orthogonal space where original convolution kernels are decomposed into principal kernels using singular value decomposition (SVD). To approximate the original server model, PriSM utilizes a novel sampling process, where a principal kernel with a larger singular value has a higher sampling probability. The probabilistic process ensures that all sub-models can together provide nearly full coverage of the principal kernels, thus reaching the near full-model training performance with significantly reduced costs on local computation and communication during sub-model aggregation. PriSM further improves memory efficiency by exploiting low-rank structure in intermediate activations and allows each client to learn only a subset of these representations while still preserving training performance. Thus, computation, memory, and communication bottlenecks at the edge are effectively resolved.

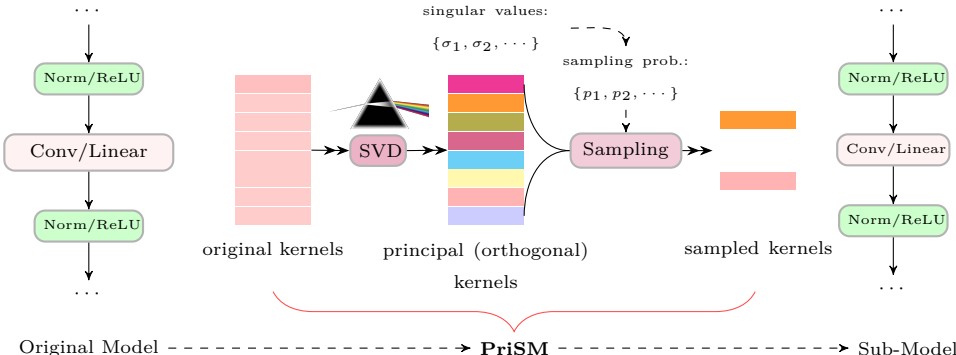

Figure 2: Creating clients' sub-models. PriSM samples a subset of principal kernels to create a client's sub-model with less computation and communication overhead. The importance-aware sampling scheme (based on singular values) ensures every sub-model approximates the full large model, and all sub-models together provide nearly full coverage of the principal kernels. PriSM further improves memory efficiency by allowing each sub-model to learn only a subset of intermediate representations.

In the following, we summarize our contributions as follows:

- We propose a novel algorithm PriSM to overcome resource constraints of large model training in federated settings. PriSM uses a novel probabilistic sampling scheme to select a subset of the kernel and create a sub-model for each resource-constrained client.

- We present comprehensive empirical evidence showing that kernel sampling is crucial to preserve models' performance, rather than simply fixed kernel dropout.

- We conduct extensive evaluations of PriSM on vision and language tasks under resourced-constrained settings where no client is capable of training the large full model. In particular, we consider both resource constraints and heterogeneity in system capacities as well as data distribution.

- We further provide detailed insights into the performance gains attained by PriSM via 1) analyzing server model's rank structure during training; 2) investigating different sampling methods; 3) profiling the kernel sampling process; 4) breaking down costs in the system.

Our results demonstrate that PriSM delivers consistently better performance compared to prior works, especially when participating clients have very limited capacities. For instance, on ResNet-18/CIFAR-10 (see Figure 5), we show that PriSM only incurs around marginal accuracy drop for i.i.d and non-i.i.d datasets under a constrained setting where clients train sub-models with only 20% of the principal kernels, accounting for $\sim 5\%$ of the full server model. Compared to other solutions, PriSM improves the accuracy by up to 10%.

## 2 Related Works

**Factorized Models**. Training neural networks with layer factorization has been extensively studied in prior literature (Denton et al., 2014; Khodak et al., 2021; Jaderberg et al., 2014; Novikov et al., 2015; Ioannou

et al., 2016; Wang et al., 2023; Niu et al., 2017; Mei et al., 2017). These works are based on the observation that well-trained neural networks have inherently low-rank structure and exhibit large correlations across kernels. A model's rank can be pre-defined or learned and adapted with layers during training (Wang et al., 2023). Hence, one can down-size the model with a low-rank approximation to provide a significant reduction in computations thus speeding up training. Furthermore, this can make model training more affordable for resource-constrained devices. In addition to models' low-rank structure, works such as (Niu et al., 2022) also exploit low-rank structure in intermediate representations to reduce computation overhead.

**Resource-Constrained Federated Learning**. While federated learning opens the door for collaborative model training over edge users having rich (but private) data, the computation, memory, and communication footprint prohibits training of large models at the resource-constrained clients. To address these resource limitations in federated learning, a number of works have been proposed in the literature (Diao et al., 2021; Horvath et al., 2021; Yao et al., 2021; Diao et al., 2021; Horvath et al., 2021; Vepakomma et al., 2018; Poirot et al., 2019; Chopra et al., 2021; He et al., 2020; Prakash et al., 2020; Elkordy et al., 2022). Particularly, in split learning (Vepakomma et al., 2018; Poirot et al., 2019; Chopra et al., 2021), the model is partitioned into two parts, one (small) part is assigned to clients for local training, while the other (large) part is outsourced to the server. He et al. (2020) proposes FedGKT that combines the model splitting approach with a bi-directional knowledge transfer technique between server and clients to achieve resource-constrained FL with much fewer communications than split learning. However, these works require sharing of intermediate activations (and in many cases, logits as well as labels) with the server, directly leaking input information and potentially costing the privacy promise of FL (Erdoğan et al., 2022; Zhang et al., 2020).

The works closely related to ours are HeteroFL (Diao et al., 2021), FjORD (Horvath et al., 2021), SplitMix (Hong et al., 2022), FLANC (Mei et al., 2022) and FedHM (Yao et al., 2021), that aim to enable participation of a resource-constrained client by letting it train a smaller sub-model based on its capabilities. In particular, HeteroFL and FjORD create sub-models for clients by selecting a certain fixed number of kernels of the server model based on each client's capacity. During model aggregation, the server aggregates each kernel and recovers the full model (if all kernels are trained). Similarly, SplitMix also splits a large model into several sub-models. However, it trains and aggregates each sub-model separately, leading to a server model as an ensemble of sub-models. On the other hand, FedHM and FLANC create sub-models using fixed subsets of factorized principal kernels. However, in these works, the size of the server model gets limited by the clients with maximum computation, memory, and communication capacities, sacrificing the model performance. In particular, methods like FedHM incur more memory footprints for intermediate representations, even for the small sub-models. This becomes even more critical in the realistic, cross-device FL setting wherein no client has the capacity to train a large model.

Structured pruning methods such as FedMask (Li et al., 2021) are also aimed at reducing training costs by pruning channels. However, FedMask requires a warm-up stage with full-model training, which does not fit into our setting that no clients can train a full model. Besides these sub-model training methods for reducing computations, FedPara (Hyeon-Woo et al., 2021), proposes a low-rank factorized model training to reduce communication costs. However computational footprint still remains prohibitive as each client conducts full-model training. FeDepth (Zhang et al., 2023) targets memory efficiency on edge clients. Instead of training a full model, it splits a full model into blocks and sequentially trains each block. As FeDepth still needs to train all blocks and upload models with the same size as the full model, computation, and communication costs are not reduced.

Therefore, to overcome the aforementioned challenges, we propose a new training method to effectively address the computation, memory, and communication bottleneck at the edge, while still preserving the privacy promises of FL.

# 3 Method

In this section, we first motivate our proposal, Principal random Sub-Model training (PriSM), with an observation of orthogonality in neural network layers. Then, we describe the details of PriSM.

**Notations**– $\|\cdot\|_F$: Frobenius norm. $\sigma_i$: $i$-th singular value in a matrix. $\circledast$: convolution. $\cdot$: matrix multiplication. $\langle\cdot,\cdot\rangle$: sum of element-wise multiplication or inner product. $tr(A)$: trace of a matrix.

### 3.1 Motivation: An Observation on Orthogonality

Without loss of generality, we consider a convolution layer with kernels $W \in \mathbb{R}^{N \times M \times k \times k}$ and input $X \in \mathbb{R}^{M \times h \times w}$, where $N$ and $M$ denote the number of output channels and input channels, $k$ is kernel size, and $h \times w$ is the size of the input image along each channel. Based on a common technique *im2col* (Chellapilla et al., 2006), the convolution layer can be converted to matrix multiplication as $\overline{Y} = \overline{W} \cdot \overline{X}$, where $\overline{W} \in \mathbb{R}^{N \times Mk^2}$ and $\overline{X} \in \mathbb{R}^{Mk^2 \times hw}$. For kernel decorrelation, we apply singular value decomposition (SVD) to map kernels into orthogonal space as: $\overline{W} = \sum_{i=1}^{N} \sigma_i \cdot \boldsymbol{u}_i \cdot \boldsymbol{v}_i^T$, where $\{\boldsymbol{u}_i\}_{i=1}^{N}$, $\{\boldsymbol{v}_i\}_{i=1}^{N}$ are two sets of orthogonal vectors[2]. The convolution can be decomposed as

$$\overline{Y} = \sum_{i=1}^{N} \overline{Y}_i = \sum_{i=1}^{N} \sigma_i \cdot \boldsymbol{u}_i \cdot \boldsymbol{v}_i^T \cdot \overline{X}. \tag{1}$$

For $\forall i \neq j$, it is easy to verify that $\langle \overline{Y}_i, \overline{Y}_j \rangle = \sigma_i \cdot \sigma_j \cdot tr(\overline{X}^T \cdot \boldsymbol{v}_i \cdot \boldsymbol{u}_i^T \cdot \boldsymbol{u}_j \cdot \boldsymbol{v}_j^T \cdot \overline{X}) = 0$, namely the output features $\overline{Y}_i$ and $\overline{Y}_j$ are orthogonal. Therefore, if we regard $\overline{W}_i = \sigma_i \cdot \boldsymbol{u}_i \cdot \boldsymbol{v}_i^T$ as a principal kernel, different principal kernels create orthogonal output features. To illustrate this, Figure 3 shows an input image (left) and the outputs (right three) generated by principal kernels. We can observe that principal kernels captures different features and serve different purposes.

As revealed in (Xie et al., 2017; Balestriero et al., 2018; Wang et al., 2020), imposing orthogonality on kernels leads to better training performance. This motivates us to initiate the training with a set of orthogonal kernels. Furthermore, to preserve kernel orthogonality during training, it is critical to constantly refresh the orthogonal space through re-decomposition. The above intuitions based on the observation on orthogonality play a key role in PriSM, which is described in the following section.

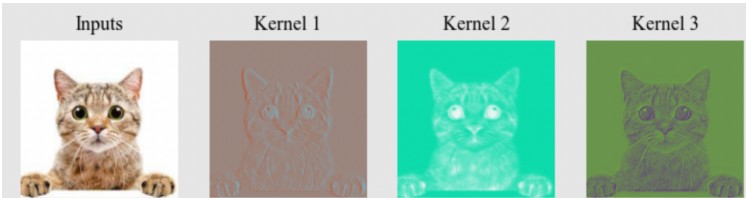

Figure 3: Orthogonal features generated by principal kernels. Different principal kernels capture different features (Kernel 1: outline of the cat, Kernel 2 and 3: textures but on distinct regions).

**Remark 3.1.** *Besides the convolution layer discussed above, the layer decomposition using SVD also applies to general linear layers (e.g., MLPs, LSTMs, Transformers), where weights $W \in \mathbb{R}^{N \times M}$. In particular, we can directly apply SVD and obtain the principal components.*

### 3.2 PriSM: Principal Probabilistic Sub-Model Training

Motivated by the observation that output features are orthogonal given orthogonal kernels, we propose PriSM, a new sub-model training method that directly trains orthogonal kernels in clients. Particularly, PriSM introduces two key components to ensure training performance with sub-models under very constrained settings. First, considering different orthogonal kernels and their contributions to outputs are weighed by the corresponding singular values, PriSM devises a novel importance-aware sampling scheme to create client sub-models to achieve computation and communication efficiency. In particular, the sampling scheme brings two benefits: i) each sub-model is a low-rank approximation of the server model; ii) the conglomerate of the sampled sub-models enables nearly full coverage of orthogonal kernels. Second, to further improve the memory efficiency while maintaining training performance, PriSM exploits low-rank structure in activations and allows each client to learn only a subset of intermediate representations in each layer. We start with unfolding sub-model creation, and then describe the training procedure in PriSM.

---

[2]We assume w.l.o.g $\overline{W}$ is a tall matrix.

### 3.2.1 Sub-Model Creation

This section explains how PriSM creates different sub-models for local clients.

**Computation and communication efficiency via sub-model sampling.** For a convolution layer with principal kernels $\left\{\overline{W}_i\right\}_{i=1}^{N}$, and the corresponding singular values $\{\sigma_i\}_{i=1}^{N}$, we randomly sample $r$ principal kernels denoted by $\overline{W}^c$ with sampling probability for $i$-th kernel as follows:

$$p_i = \frac{\sigma_i^{\kappa}}{\sum_{j=1}^{N} \sigma_j^{\kappa}}. \tag{2}$$

Here, for a given layer, $r$ is decided by $c$-th client's resource budget, and $\kappa$ is a smooth factor controlling the probability distribution in sampling. Indices of selected kernels are denoted as $\mathcal{I}_c$.

The convolution output is calculated as

$$\overline{Y} = \sum_{i \in \mathcal{I}_c} \overline{Y}_i = \sum_{i \in \mathcal{I}_c} \sigma_i \cdot \boldsymbol{u}_i \cdot \boldsymbol{v}_i^T \cdot \overline{X}. \tag{3}$$

By performing the sampling process for every convolution layer, we create a *random* low-rank model for each client. *Important* kernels with large singular values are more likely to be chosen, and all sub-models can together provide nearly full coverage of principal kernels. The resulting sub-model hence has convolution layers with fewer kernels, reducing required computations and communication overhead (when aggregating models). Other element-wise layers, such as ReLU and batch normalization, remain the same.

**Memory efficiency via learning a subset of features.** We further exploit low-rank structure in intermediate representations. For a sub-model above, each convolution layer is decomposed into two sublayers: $Conv_U$ and $Conv_V$ with kernels $\{\boldsymbol{u}_i\}_{i \in \mathcal{I}_c}$ and $\{\boldsymbol{v}_i\}_{i \in \mathcal{I}_c}$ respectively. Without further optimization, output from $Conv_U$ consumes the same memory as the original layer, causing high memory pressure at the edge. However, inputs to $Conv_U$, $\overline{X} \in \mathbb{R}^{r \times hw}$, and kernels $\overline{W} \in \mathbb{R}^{N \times r}$ are both low-rank. Therefore, based on the rank inequality of matrix multiplication (Banerjee & Roy, 2014), we obtain

$$\mathrm{Rank}(\overline{Y}) \leq \min\left(\mathrm{Rank}(\overline{X}), \mathrm{Rank}(\overline{W})\right) \leq r. \tag{4}$$

Therefore, the output of $Conv_U$ is also low-rank. Such a low-rank structure reflects correlation among output channels, indicating channel redundancy in outputs when $r < N$. Based on the observation, we further allow $Conv_U$ to compute only a subset of output channels, therefore reducing memory footprints of activations while still preserving necessary information. Typically, given input to $Conv_U$ with $r$ input channels, PriSM only computes $r$ output channels. During implementation, we replace the select $\boldsymbol{u}_i$ with its subset $\boldsymbol{u}_i(1:r)$ when computing output features from $Conv_U$. For simplicity, we will still use $\boldsymbol{u}_i$ in the rest of the paper to denote the subset $\boldsymbol{u}_i(1:r)$. The unselected channels are cached in $\hat{\boldsymbol{u}}_i$, and are used in model aggregation.

### 3.2.2 Training

This section details the training procedure in PriSM. We describe each component below.

**Local training.** On each client, during local training, the sub-model with parameter $\{\sigma_i, \boldsymbol{u}_i, \boldsymbol{v}_i\}_{i \in \mathcal{I}_c}$ are updated. The sub-model also consists of trainable layers such as BatchNorm but with fewer channels denoted in $\mathcal{I}_c$. PriSM allows $\sigma_i$ to be trained in local training, thus ensuring changes regarding each principal kernel's importance are captured in local clients. In addition, $\sigma_i$ will be merged into $\boldsymbol{u}_i, \boldsymbol{v}_i$ as $\boldsymbol{u}_i' = \sqrt{\sigma_i}\boldsymbol{u}_i, \boldsymbol{v}_i' = \sqrt{\sigma_i}\boldsymbol{v}_i$ to reduce memory footprints.

**Sub-model aggregation.** On the server side, with sub-models obtained from clients, we aggregate $i$-th principal kernel as follows:

$$\overline{W}_i = \left(\left[\sum_{c \in \mathcal{C}} \alpha_i^c \boldsymbol{u}_i'^c; \quad \hat{\boldsymbol{u}}_i\right]\right) \cdot \left(\sum_{c \in \mathcal{C}} \alpha_i^c \boldsymbol{v}_i'^c\right)^T, \tag{5}$$

where $\mathcal{C}$ denotes the subset of active clients, $\alpha_i$ is the aggregation coefficient for $i$-th kernel. We propose a weighted averaging scheme: if $i$-th kernel is selected and trained by $C_i$ clients, then $\alpha_i^c = 1/C_i$. Furthermore, the unselected output channels $\hat{\boldsymbol{u}}_i$ are concatenated with the aggregated $\boldsymbol{u}_i$ before reconstructing the orthogonal kernel to preserve model capacity. If $\overline{W}_i$ was not selected by any client, it remained unchanged in the server. The full model in the original space is constructed by converting each 2-dimensional $\overline{W}_i$ to the original dimension $\mathbb{R}^{M \times k \times k}$ and combining them.

**Orthogonal space refresh.** After model aggregation, we perform SVD on the updated kernels $\overline{W}$ to preserve orthogonality among the principal kernels. Thus, in the next communication round, the importance-aware sampling can still create low-rank sub-models for different clients..

We further use two additional techniques to improve learning efficiency in the orthogonal space: activation normalization, and regularization on orthogonal kernels.

**Activation normalization.** We apply batch normalization without tracking running statistics; namely, the normalization always uses current batch statistics in the training and evaluation phases. Each client applies normalization separately with no sharing of statistics during model aggregation. Such an adaptation is effective in ensuring consistent outputs between different sub-models and avoids potential privacy leakage through the running statistics (Andreux et al., 2020).

**Regularization.** When learning a factorized model on a client, applying weight decay to $\boldsymbol{u}_i$ and $\boldsymbol{v}_i$ separately results in poor final accuracy. Inspired by Khodak et al. (2020), for training on client $c$, we add regularization to the subset of kernels as follows:

$$reg = \frac{\lambda}{2} \left\| \sum_{i \in \mathcal{I}_c} \boldsymbol{u}'_i \cdot \boldsymbol{v}'^{T}_i \right\|_F^2, \tag{6}$$

where $\lambda$ is the regularization factor, $\mathcal{I}_c$ denotes the subset of principal kernels on client $c$.

Algorithm 1 presents an overall description of PriSM. We only show the procedure on a single convolution layer with kernels $W$ for the sake of simplifying notations.

---

**Algorithm 1** PriSM: Principal Probabilistic Sub-Model Training

---

**Input:** layer parameters $W$, client capacities.

1: Decompose $W$ into orthogonal kernel using SVD $\rightarrow \{\overline{W}_i\}_{i=1}^N$.
2: **for** communication round $t = 1, \cdots, T$ **do**
3:     Choose a subset of clients $\rightarrow \mathcal{C}$.
4:     **for** each client $c \in \mathcal{C}$ **do**
5:         Compute the sub-model size for client $c \rightarrow |\mathcal{I}_c|$.
6:         Sub-model: Obtain a sub-model following the procedure in Sec 3.2.1 $\rightarrow \mathcal{I}_c, \overline{W}^c$.
7:         Local training: Perform **LocalTrain** $\leftrightarrow \mathcal{I}_c, \overline{W}^c$.
8:     **end for**
9:     Sub-model aggregation: Aggregate parameters based on Eq. (5) $\overline{W} \leftarrow \{\overline{W}^c\}_{c \in \mathcal{C}}$.
10:     Orthogonal space refresh: Perform SVD on $\overline{W}$.
11: **end for**

---

12: **LocalTrain** $\leftrightarrow \mathcal{I}_c, \overline{W}^c$
13: **for** local iteration $k = 1, \cdots, K$ **do**
14:     Sample an input batch from the local dataset $\rightarrow \mathcal{D}_k$.
15:     Perform the forward and backward pass $\leftarrow \mathcal{D}_k, \overline{W}^c$.
16:     Update the local sub-model using SGD $\rightarrow \overline{W}^c$.
17: **end for**

---

In the following remarks, we differentiate PriSM from Dropout and Low-Rank compression.

**Remark 3.2.** ***PriSM vs Dropout.*** *PriSM shares some computation similarity with model training using regular dropout (Diao et al., 2021; Horvath et al., 2021). However, methods with regular dropout simply select a fixed subset of kernels with size depending on a client's resource constraints, and suffer severe performance degradation, esp. with a high dropout probability. In contrast, PriSM performs importance-aware sampling in the orthogonal space. Each sub-model approximates the full model, and different sub-models do not create significant inconsistency. Importantly, PriSM obtains a final global model with full capacity, even though only allows clients to train sub-models.*

**Remark 3.3.** ***PriSM vs Low-Rank Compression.*** *PriSM is not a low-rank compression method. Low-rank compression methods such as FedHM (Yao et al., 2021) aim to construct a smaller server model by completely discarding some kernels even though they can still contribute to training performance (See Figure 1 in Section 1 and Figure 8 in Section 4.4). PriSM randomly select sub-models so that every kernel is possible to be learned. Furthermore, PriSM achieves memory efficiency by exploiting low-rank properties in intermediate representations, which is not seen in other low-rank methods.*

## 4  Experiments

In this section, we present numerical analyses of PriSM. First, we study the necessity of probabilistic kernel sampling in preserving models' capacity and accuracy via centralized training. Then, we evaluate PriSM under resource-constrained settings where no clients can train the large full model. Furthermore, we consider both homogeneous and heterogeneous client settings. Specifically, in homogeneous settings, all clients have the same limited computation, memory, and communication capacity, while in heterogeneous settings, clients' capacities might vary. We also compare PriSM with three other baselines: ordered dropout in orthogonal space (OrthDrop), ordered dropout in original space (OrigDrop), and SplitMix (Hong et al., 2022). At a high level, our results demonstrate that PriSM achieves comparable accuracy to full-model training even when only training very small sub-models on all clients. Additionally, we provide more insights into the superior performance of PriSM by 1) analyzing the server model's rank structure; 2) profiling the kernel sampling process during training; 3) investigating cost breakdowns in a federated system. Finally, we conduct ablation studies including analyzing several potential sampling strategies and impacts of training hyperparameters.

**Baselines.** Prior methods such as FjORD (Horvath et al., 2021) and HeteroFL (Diao et al., 2021) select sub-models from the original kernel space, for which we denote as OrigDrop. In implementation, we follow the procedure in HeteroFL. On the other hand, we use OrthDrop to denote selecting fixed top-k kernels from the orthogonal space such as in FedHM (Yao et al., 2021). For SplitMix, we follow the procedure in (Hong et al., 2022) to create atom sub-models and evaluate the ensemble of all sub-models on the server side.

**Models and Datasets.** We train ResNet-18 on CIFAR-10 (Krizhevsky et al., 2009), CNN on FEMNIST (Caldas et al., 2018) and LSTM model on IMDB (Maas et al., 2011). We also train a transformer model DeiT(Touvron et al., 2021) on CIFAR-10. Detailed architectures are provided in Appendix A.1.

**Data Distribution.** For CIFAR-10 and IMDB, we uniformly sample an equal number of images for each client when creating i.i.d datasets. For non-i.i.d datasets, we first use Dirichlet function $\text{Dir}(\alpha)$ (Reddi et al., 2020) to create sampling probability for each client and then sample an equal number of training images for clients. We create two different non-i.i.d datasets with $\alpha = 1$ and $\alpha = 0.1$, where a smaller $\alpha$ indicates a higher degree of non-i.i.d. For FEMNIST, we directly use the dataset without any additional preprocessing.

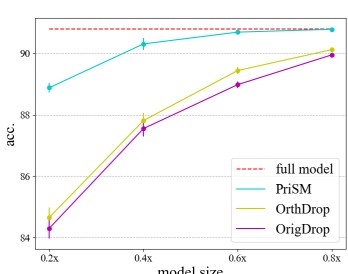

Figure 4: Accuracy of VGG11 on CIFAR-10 with different sub-models. PriSM delivers much better performance under very constrained settings (e.g., training with only 0.2× sub-models) compared to OrthDrop and OrigDrop.

**FL Setting.** We simulate an FL setting with 100 clients with 20 random clients active in each communication round. Each client trains its model for 2 local epochs in each round. We use SGD with momentum during training. The learning rate is initially 0.1 for ResNet-18 and DeiT and 0.01 for CNN/FEMNIST, and decayed by a cosine annealing schedule. Further details are in Appendix A.2.

## 4.1 Probabilistic Sub-Model Training in Centralized Settings

We first demonstrate that probabilistic kernel sampling is essential to preserve models' capacity and maintain accuracy in centralized settings. While centralized training is not our goal, it provides direct evidence that PriSM reduces training complexities without significantly sacrificing accuracy.

We use VGG11 as an example and train the model on CIFAR-10. During training, in each iteration, we sample a subset of kernels and create a sub-model as described in PriSM. In different iterations, different subsets of kernels will be sampled and trained. We adopt the following hyperparameters during training( SGD, *batch size*=128, *weight decay*=1e-4, *epochs*=150). To further demonstrate the benefits of kernel sampling, we also compared the accuracy with OrigDrop and OrthDrop, where we respectively extract a fixed subset of kernels and create sub-models in the original and orthogonal kernel space.

Figure 4 shows the accuracy of VGG11 on CIFAR-10 with different sub-models. We observe that PriSM achieves much higher accuracy compared to the other two baselines, especially under very constrained settings (e.g., $0.2\times$ sub-models). Hence, compared to these compression methods that reduce models' capacity, PriSM preserves models' capacity via the probabilistic kernel sampling, therefore maintaining model accuracy.

## 4.2 Performance on Constrained Homogeneous Clients

Table 1: Model size, MACs, activation memory for sub-models in PriSM (batch size: 32). Unlike conventional low-rank methods, PriSM greatly reduces model size, computations, and activation memory when training with very small sub-models.

| Model | | Full | $0.8\times$ | $0.6\times$ | $0.4\times$ | $0.2\times$ |
|---|---|---|---|---|---|---|
| ResNet-18 | | | | PriSM | | |
| | Params | 11 M | 7.9 M (72%) | 4.5 M (41%) | 2 M (18%) | .5 M (4.5%) |
| | MACs | 35 G | 25.5 G (73%) | 14.7 G (42%) | 6.87 G (20%) | 2 G (5.6%) |
| | Mem | 31.5 M | 37 M (115%) | 28.3 M (90%) | 18.9 M (60%) | 9.5 M (30%) |
| | | | | OrthDrop (FedHM (Yao et al., 2021)) | | |
| | Params | 11 M | 9.9 M (90%) | 7.4 M (67%) | 4.9 M (44%) | 2.5 M (22%) |
| | MACs | 35 G | 28.8 G (82%) | 22.4 G (64%) | 16 G (46%) | 7.4 G (23%) |
| | Mem | 31.5 M | 44.9 M (143%) | 40.5 M (129%) | 36.1 M (115%) | 31.7 M (101%) |

In this setting, we assume that all clients have the same limited capacity. We vary the client sub-model size from 0.2 to 0.8 of the full server model, where $0.x$ indicates that only a $0.x$ subset of the principal kernels are sampled in each convolution layer from the server model (denoted as *keep ratio*). Accordingly, for a given layer, $r = 0.x \times N$, where $N$ is the number of output channels in the layer. In Table 1, we list the computation, memory and communication footprints for sub-models of ResNet-18. DeiT/CIFAR-10, CNN/FEMNIST, and LSTM/IMDB also enjoy similar cost reductions. It is worth noting that PriSM incurs much smaller costs compared to the deterministic compression methods(FedHM), thanks to the memory-efficiency design.

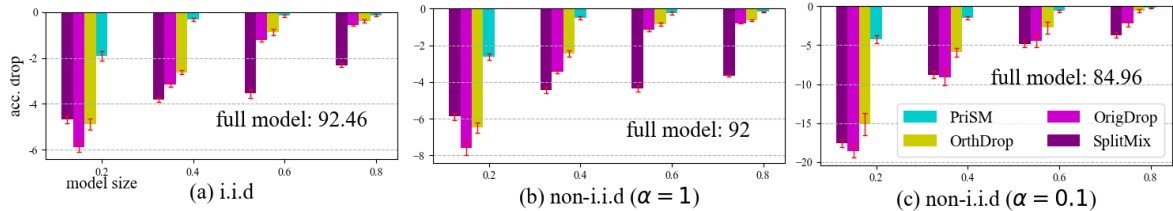

Figure 5: Accuracy drops on ResNet18/CIFAR-10 on homogeneous clients compared to full-model training. PriSM constantly delivers better performance compared to OrigDrop, OrthDrop, and SplitMix, significantly outperforming them under very constrained realistic edge settings. (bar: mean; line: std)

**ResNet-18/CIFAR-10**. Figure 5 shows final validation accuracy drops of ResNet-18 with different sub-models on i.i.d and non-i.i.d datasets compared to full-model training. We note that PriSM constantly

delivers better performance than the other three baselines. The performance gap is more striking under very constrained settings. For instance, when only 0.2× sub-models are supported on clients, PriSM attains comparable accuracy as full-model training, and achieves up to 10%/14%/13% performance improvement compared to OrthDrop, OrigDrop and SplitMix on non-i.i.d dataset with $\alpha = 0.1$. We also make two key observations. First, training with sub-models in the orthogonal space provides better performance than in the original space, which aligns with our intuition in Section 3.1. Second, our importance-aware sampling strategy for creating sub-models is indispensable as demonstrated by the notable performance gap between PriSM, OrthDrop and SplitMix.

**CNN/FEMNIST**. We adopt a CNN model as in (Horvath et al., 2021), which consists of two convolution layers (See Table 8 in Appendix for details). During training, we simulate 100 clients, in which each client contains 10 users' data from FEMNIST (Caldas et al., 2018). Figure 6 shows accuracy drops of sub-model training with different sizes. Similar as in the ResNet experiments, PriSM incurs very small accuracy drops even under very constrained settings. In particular, 0.2× sub-model training only results in less than 1% accuracy drops, greatly outperforming the other three baselines.

**LSTM/IMDB**. The LSTM model used in FL training is detailed in Table 8. During training, we simulate 100 clients, in which each client is assigned 375 training samples. We create local datasets with two distributions using the same method as in CIFAR-10: i.i.d and non-i.i.d ($\alpha = 0.1$). Table 12 (in Appendix) lists detailed training hyperparameters. Figure 7 shows accuracy drops of sub-model training on IMDB in homogeneous client settings. While the task on IMDB is just a binary classification problem, PriSM still achieves the best final server model accuracy on both i.i.d and non-i.i.d datasets. On i.i.d datasets, training using sub-models achieves comparable accuracy as full-model training, even only using very small sub-models such as 0.2×. On the other hand, on non-i.i.d dataset, OrigDrop suffers notable accuracy drops compared to PriSM.

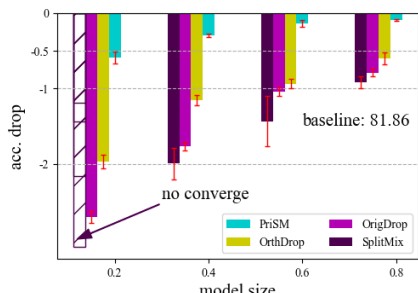

Figure 6: Accuracy drops on FEMNIST on homogeneous clients compared to full-model training. PriSM incurs smaller accuracy drops under very constrained settings than the other three baselines.

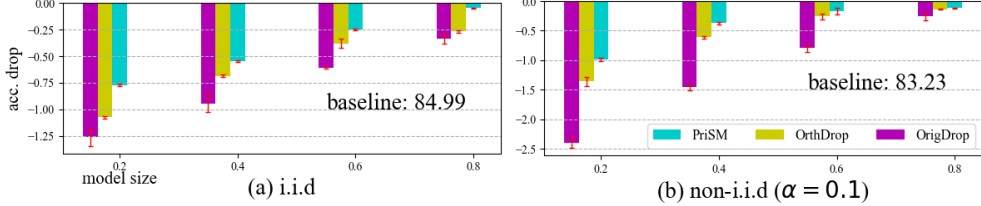

Figure 7: Accuracy drops on IMDB on homogeneous clients compared to full-model training.

**Transformer/CIFAR-10**. PriSM also applies to the transformer architecture. To that end, we test PriSM on DeiT (Touvron et al., 2021), an efficient image transformer model. It consists of 12 attention layer, and each layer has 3 attention heads. Detailed architecture is provided in Table 6 (in Appendix A.1). We use a pre-trained model on ImageNet and adopt the same hyperparameter settings as in ResNet-18/CIFAR-10. Table 2 shows the final global model performance with different sub-model training. Similar to ResNet-18/CIFAR-10, PriSM maintains comparable accuracy on both i.i.d and non-i.i.d datasets even under very constrained settings such as when clients can only train 0.2× or 0.4× sub-models. Therefore, thanks to the probabilistic sampling process and memory-efficiency design, PriSM preserves the server model's capacity while significantly reducing training costs.

Table 2: Training performance of DeiT/CIFAR-10 resource-constrained clients.

| Distribution | Full Model (Baseline) | 0.8× | 0.6× | 0.4× | 0.2× |
|---|---|---|---|---|---|
| i.i.d | 92.04 | $91.92 \pm 0.18$ | $91.36 \pm 0.26$ | $90.12 \pm 0.31$ | $88.34 \pm 0.28$ |
| non-i.i.d ($\alpha = 1$) | 91.14 | $90.84 \pm 0.15$ | $90.50 \pm 0.18$ | $89.72 \pm 0.21$ | $87.87 \pm 0.32$ |

### 4.3 Performance on Constrained Heterogeneous Clients

To simulate clients with varying limited capacity, we simulate the following setting: 40% clients train 0.4× sub-models, and 60% clients train 0.2× sub-models. No participating client trains the full model. For baseline methods, we follow the same strategy as in Section 4.2.

Table 3 lists the final accuracy achieved by different methods under the heterogeneous setting. PriSM greatly outperforms the baseline methods even when 0.4× sub-models are supported on a small fraction of clients. Furthermore, similar to the results in Section 4.2, the benefits of training in the orthogonal space and importance-aware sampling strategy are also observed in heterogeneous client settings.

Table 3: Training performance of ResNet-18/CIFAR-10 on heterogeneous clients.

| Distribution | Full Model (Baseline) | OrigDrop | OrthDrop | PriSM |
|---|---|---|---|---|
| i.i.d | 92.46 | $88.86 \pm 0.17$ | $89.57 \pm 0.16$ | $90.63 \pm 0.14$ |
| non-i.i.d (1) | 92 | $86.91 \pm 0.24$ | $88.78 \pm 0.22$ | $89.89 \pm 0.2$ |
| non-i.i.d (0.1) | 84.96 | $76.38 \pm 0.92$ | $78.37 \pm 0.99$ | $82.58 \pm 0.51$ |

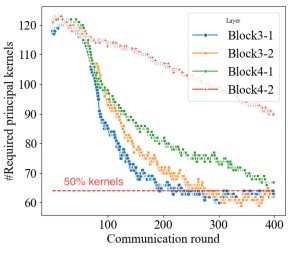
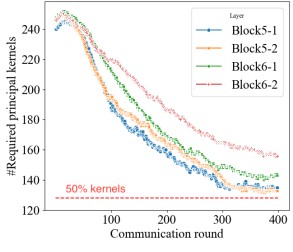
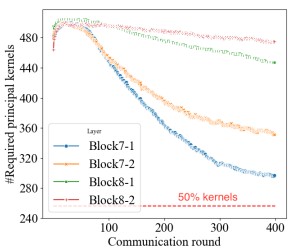

(a) ResBlock 3, 4 (128 kernels).    (b) ResBlock 5, 6 (256 kernels).    (c) ResBlock 7, 8 (512 kernels).

Figure 8: The number of principal kernels required to accurately approximate each convolution layer in ResBlocks 3-8 in ResNet-18 (Results of ResBlocks 1 and 2 are discussed in Figure 1). The server model gradually attains a low-rank structure. However, even in the final model, half of the principal kernels are still required to approximate most layers.

### 4.4 Insights into PriSM

We now focus on providing further insights into PriSM by analyzing some of its key aspects. To this end, we first examine the low-rank structure of models during training, and pinpoint the cause behind the accuracy gap between fixed and random kernel dropout strategies in the orthogonal space. Thereafter, we study the sampling process and cost breakdown in the FL system

#### 4.4.1 Model's rank during training.

To analyze the server model's rank structure, we adopt a similar method as in Alter et al. (2000) to calculate the required number of principal kernels to accurately approximate each layer as $2^{-\log\left(\sum_i p_i \log p_i\right)}$. Here, $p_i$ is calculated as in Eq. (2) with $\kappa = 1$. Figure 8 shows the number of required kernels for each layer in ResNet-18 during full-model FL (Block$i$-$j$: $j$-th convolution layer in $i$-th ResBlock). We observe that a randomly initialized model is not low-rank, rather the server model attains a low-rank structure gradually. Therefore, sub-models with fixed top-k principal kernels inevitably cause reductions in the server model capacity. Furthermore, even at the end of the training, around half the principal kernels are still required to approximate most layers. In fact, some layers require even more principal kernels. Therefore, our probabilistic sampling scheme is essential in preserving the server model capacity during FL training with sub-models.

#### 4.4.2 Kernel sampling profiling.

Figure 9 shows the average number of clients assigned for each orthogonal channel in one communication round. Each client trains a 0.2× sub-model of ResNet-18 on CIFAR-10 with i.i.d distributions as in Sec 4.2. We observe that each kernel is selected by at least one client in each round, indicating every kernel will be activated and trained on clients in each round. Furthermore, orthogonal kernels with larger $\sigma$ get more chances to be chosen, which ensures sub-models on all clients consistently approximate the full model.

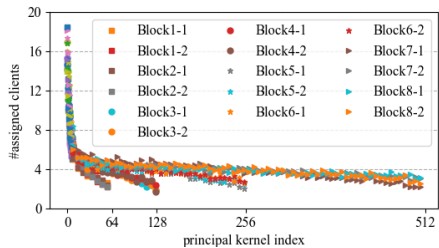

Figure 9: Average number of clients assigned for each orth. kernel during training. Kernels with large singular values get more chances to be chosen.

### 4.4.3 Runtime breakdown.

In this section, we investigate the relative cost of SVD by breaking down the model training time into four stages: sub-model creation, local training, model aggregation, and obtaining orthogonal kernels (SVD). We choose a large model, ResNet-18, as the target model. We adopt hyperparameters as in Table 10. We run the server and client process on NVIDIA RTX 5000. Table 4 lists each stage's average time in one communication round. We observe that the time of SVD is nearly negligible compared to the time spent on local training. Therefore, even for large models, the relative overhead of SVD is still very small.

Table 4: Training time breakdown for ResNet-18 on CIFAR-10.

| stage | sub-model create | local train | aggregate | SVD |
|---|---|---|---|---|
| time | 3.27 s | 36.82 s | 0.11 s | 0.96 s |

## 4.5 Ablation Study: Different Sampling Methods

We also conducted an additional study on the effects of different sampling strategies on the final serve model accuracy. In addition to the sampling method in PriSM (Eq (2)), we also try two other methods: `uniform` sampling and `softmax-based` sampling. In `uniform sampling`, we regard each channel as equally important with a uniform sampling probability. On the other hand, the `softmax-based` sampling assigns each channel a probability that is calculated by applying the softmax function to the singular values. Hence, it also assigns high sampling probabilities for channels with large singular values.

Table 5 shows the final validation accuracy of the global model. We observe that training with `uniform` sampling incurs significant accuracy drops in both i.i.d and non-i.i.d data distributions. It indicates that importance-aware sampling is crucial to avoid a large variation between sub-models and the full model. On the other hand, while the `softmax-based` sampling also considers each kernel's importance, the sampling method in PriSM performs better in choosing the right sub-model and covering the whole kernel space.

We also study the effects of two hyperparameters: the number of local epochs and active clients. Each of these two parameters affects the sampling process when creating sub-models for clients. Specifically, given a fixed number of total iterations, FL training with a small number of local epochs per communication round performs a more frequent sampling process, thus making more orthogonal kernels to be selected and trained. Similarly, the FL training with a large number of active clients per round can also activate more orthogonal kernels. Due to space constraints, the results are deferred to Appendix A.3. At a high level, fewer local epochs (given fixed total iterations) or more active clients in each round result in higher sampling probability for each kernel. Therefore, training performance is improved as shown in the results.

Table 5: Global model accuracy with different sampling strategies.

| sampling | i.i.d | non-i.i.d ($\alpha = 1$) | non-i.i.d ($\alpha = 1$) |
|---|---|---|---|
| `uniform` | $69.54 \pm 0.21$ | $67.28 \pm 0.51$ | $57.5 \pm 0.67$ |
| `softmax-based` | $84.74 \pm 0.49$ | $83.19 \pm 0.43$ | $72.69 \pm 0.62$ |
| PriSM | $90.57 \pm 0.25$ | $89.36 \pm 0.21$ | $80.72 \pm 0.59$ |

## 5 Conclusion

We have considered a practical yet under-explored problem of federated learning in a resource-constrained edge setting, where no participating client has the capacity to train a large model. As our main contribution, we propose the PriSM training methodology, that empowers resource-limited clients by enabling them to train smaller sub-models, while still allowing the server to reconstruct the full model. Importantly, PriSM utilizes a novel sampling approach to obtain sub-models for the clients, all of which together provide near-full model coverage. PriSM further improves memory efficiency by exploiting low-rank structure in intermediate activations. Our extensive empirical results on diverse models and datasets demonstrate that PriSM performs significantly better than the prior baselines, especially when each client can train only a very small sub-model. For instance, compared to full-model training, PriSM only incurs $\sim 2\%$ accuracy drop for ResNet-18/CIFAR-10 when clients train only $0.2\times$ sub-models, while yielding an accuracy improvement of up to $10\%$ compared to existing works. We further present insights into performance gains of PriSM compared to other baseline methods and demonstrate the necessity of importance-aware kernel sampling in sub-model training.

## Acknowledgement

This material is based upon work supported by ONR grant N00014-23-1-2191, ARO grant W911NF-22-1-0165, Defense Advanced Research Projects Agency (DARPA) under Contract No. FASTNICS HR001120C0088 and HR001120C0160, and gifts from Intel and Qualcomm. The views, opinions, and/or findings expressed are those of the author(s) and should not be interpreted as representing the official views or policies of the Department of Defense or the U.S. Government.

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

# A   Models and Hyperparameters

In this section, we provide detailed information about models and hyperparameter settings for the results presented in the paper. We will open our source code upon acceptance of the paper.

## A.1   Models

**ResNet-18/CIFAR-10.**  We use a ResNet-18 optimized for CIFAR-10, in which kernel size in the first convolution layer is changed from $7 \times 7$ to $3 \times 3$. Details are shown in Table 9.

**DeiT/CIFAR-10.**  We adopt DeiT-tiny as in (Touvron et al., 2021). The detailed model is shown in Table 6.

Table 6: DeiT/CIFAR-10

| Module | kernel size | #kernel | #layers | #heads |
|---|---|---|---|---|
| Embedding | 4 | 192 | - | - |
| Attention | - | - | 12 | 3 |

**CNN/FEMNIST.**  We use a similar architecture as in FjORD Horvath et al. (2021). The detailed model is shown in Table 7.

Table 7: CNN/FEMNIST

| Module | #kernels | size | stride | ReLU |
|---|---|---|---|---|
| Conv1 | 64 | 5 | 1 | ✓ |
| Pooling1 | - | 2 | 2 | ✗ |
| Conv12 | 64 | 3 | 1 | ✓ |
| Pooling2 | - | 2 | 2 | ✗ |
| Classification | 10 | - | - | ✗ |

**LSTM/IMDB.**  We use a common LSTM model as shown in Table 8.

Table 8: LSTM/IMDB

| Module | input size | output size | hidden size | #layers |
|---|---|---|---|---|
| Embedding | 1001 | 64 | - | - |
| LSTM | 64 | 256 | 256 | 2 |
| FC | 256 | 1 | - | - |

## A.2   Training Hyperparameters

**ResNet-18,DeiT/CIFAR-10 on homogeneous clients.**  We simulate 100 clients during FL training, in which each client is assigned 500 training samples for both i.i.d and non-i.i.d datasets. In each communication round, each client performs local training for 2 epochs using the local data, then uploads parameters to the server for aggregation. Table 10 lists detailed hyperparameters during FL training with ResNet-18. For PriSM and baseline methods (OrigDrop and OrthDrop), we tune the initial learning rate from 0.01 to 0.2, with different learning rate schedulers. A cosine annealing scheduler with an initial learning rate of 0.1 gives the best performance. We keep other parameters fixed (e.g., #clients, #local epochs, #rounds, batch size) to ensure they are conducted in the same fair setting.

Table 9: ResNet-18/CIFAR-10

| Module | | #kernels | size | stride | Batch Norm | ReLU | Downsample |
|---|---|---|---|---|---|---|---|
| Conv1 | | 64 | 3 | 1 | ✓ | ✓ | ✗ |
| ResBlock 1 | Block1-1 | 64 | 3 | 1 | ✓ | ✓ | ✗ |
| | Block1-2 | 64 | 3 | 1 | ✓ | ✓ | |
| ResBlock 2 | Block2-1 | 64 | 3 | 1 | ✓ | ✓ | ✗ |
| | Block2-2 | 64 | 3 | 1 | ✓ | ✓ | |
| ResBlock 3 | Block3-1 | 128 | 3 | 1 | ✓ | ✓ | ✓ |
| | Block3-2 | 128 | 3 | 1 | ✓ | ✓ | |
| ResBlock 4 | Block4-1 | 128 | 3 | 1 | ✓ | ✓ | ✗ |
| | Block4-2 | 128 | 3 | 1 | ✓ | ✓ | |
| ResBlock 5 | Block5-1 | 256 | 3 | 1 | ✓ | ✓ | ✓ |
| | Block5-2 | 256 | 3 | 1 | ✓ | ✓ | |
| ResBlock 6 | Block6-1 | 256 | 3 | 1 | ✓ | ✓ | ✗ |
| | Block6-2 | 256 | 3 | 1 | ✓ | ✓ | |
| ResBlock 7 | Block7-1 | 512 | 3 | 1 | ✓ | ✓ | ✓ |
| | Block7-2 | 512 | 3 | 1 | ✓ | ✓ | |
| ResBlock 8 | Block8-1 | 512 | 3 | 1 | ✓ | ✓ | ✗ |
| | Block8-2 | 512 | 3 | 1 | ✓ | ✓ | |
| Classification | | 10 | - | - | ✗ | ✗ | ✗ |

Table 10: Hyperparameters for ResNet-18,DeiT/CIFAR-10 on homogeneous clients

| Datasets | #clients | #samples | distribution | augmentation |
|---|---|---|---|---|
| | 100 | 500 | i.i.d, non-i.i.d ($\alpha = 1, 0.1$) | flip, random crop |
| Training | #Rounds | #local epochs | batch size | #active clients | smooth factor $\kappa$ |
| | 1000/400 (DeiT) | 2 | 32 | 20 | 2.5/4 (0.2 sub-model) |
| Opt | Optimizer | Momentum | $wd$ | initial $lr$ | scheduler |
| | SGD | 0.9 | 0.0002 | 0.1 | cosine annealing |

**CNN/FEMNIST on homogeneous clients.** We simulate 100 clients during FL training, in which each client is assigned 10 users' data from the original training dataset. We use the whole validation dataset to compute the validation accuracy. Table 11 lists detailed hyperparameters during FL training with CNN.

Table 11: Hyperparameters for CNN/FEMNIST on homogeneous clients

| Datasets | #clients | #users/client | distribution | augmentation |
|---|---|---|---|---|
| | 100 | 10 | natural non-i.i.d | None |
| Training | #Rounds | #local epochs | batch size | #active clients | smooth factor $\kappa$ |
| | 100 | 2 | 32 | 20 | 2.5 |
| Opt | Optimizer | Momentum | $wd$ | initial $lr$ | scheduler |
| | SGD | 0.9 | 0.0002 | 0.01 | cosine annealing |

**ResNet-18/CIFAR-10 on heterogeneous clients.** We adopt the same setting as in Table 10, except the fact that clients might vary in computation and communication capacity. Therefore different model might train sub-models with different sizes (See Sec 4.3 in the main paper).

**LSTM/IMDB on homogeneous clients** The LSTM model used in FL training is detailed in Table 8. During training, we simulate 100 clients, in which each client is assigned 375 training samples. We create local datasets with two different distributions using the same method as in CIFAR-10: i.i.d and non-i.i.d ($\alpha = 0.1$). Table 12 list detailed hyperparameters for training LSTM/IMDB.

Table 12: Hyperparameters for LSTM/IMDB on homogeneous clients

| Datasets | #clients | #samples | distribution | augmentation |
|---|---|---|---|---|
| | 100 | 375 | i.i.d, non-i.i.d ($\alpha = 0.1$) | None |
| Training | #Rounds | #local epochs | batch size | #active clients | smooth factor $\kappa$ |
| | 300 | 2 | 32 | 20 | 2 |
| Opt | Optimizer | Momentum | $wd$ | initial $lr$ | scheduler |
| | SGD | 0.9 | 0.0002 | 0.1 | cosine annealing |

### A.3 More Ablation Study

### A.3.1 Effects of the number of local epochs.

To investigate the effects of the number of local epochs on the final server model accuracy, we train ResNet-18 in homogeneous client settings. The sub-model trained on clients varies from 0.2× to 0.6×. We also train a full model as a baseline. The common training hyperparameters are the same as in Table 10. We fix the number of total iterations as 2000, namely, Round×Local epochs = 2000. Figure 10 shows final server model accuracy on i.i.d and non-i.i.d datasets with $\alpha = 0.1$. We observe that while final server model accuracy decreases as the number of local epochs increases, the accuracy gap between sub-model and full-model training also slightly increases. One potential reason is that the total number of sampling decreases in FL training with more local epochs. As a result, some orthogonal kernels are under-trained. Such observation also aligns with the intuition discussed in the main paper that all orthogonal kernels should be trained.

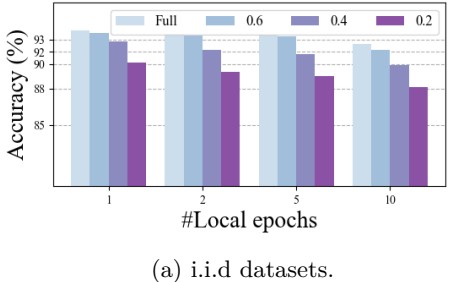
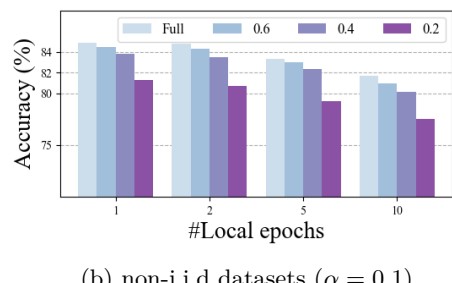

(a) i.i.d datasets.      (b) non-i.i.d datasets ($\alpha = 0.1$).

Figure 10: Effects of the number of local epochs on training performance (ResNet-18/CIFAR-10).

### A.3.2 Effects of the number of active clients.

We follow the same settings as in Table 10 except that the number of active clients varies from 10 to 30 in each communication round. Similar to above, we vary the sub-model trained on clients from 0.2× to 0.6×. Figure 11 shows the final server model accuracy on i.i.d and non-i.i.d datasets with $\alpha = 0.1$. On i.i.d datasets, FL training with a different number of active clients does not significantly change the training

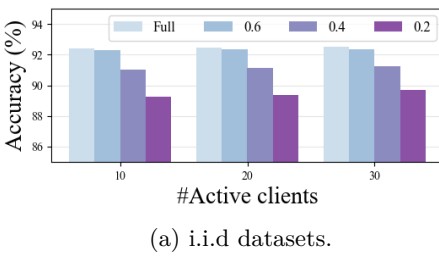
(a) i.i.d datasets.

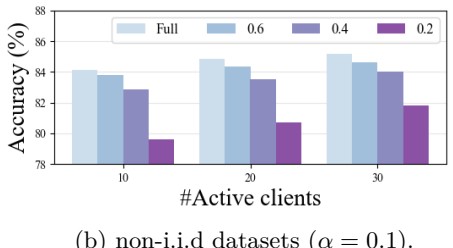
(b) non-i.i.d datasets ($\alpha = 0.1$).

Figure 11: Effects of the number of active clients on training performance (ResNet-18/CIFAR-10).

performance. Furthermore, the accuracy gap between sub-model and full-model training is also almost the same, even though a less frequent sampling process is performed in training with fewer clients. However, on non-i.i.d datasets, the final accuracy increases with an increasing number of participating clients. Further, the performance gap between sub-model and full-model training also shrinks as more sampling processes are performed in each round. In practical edge settings, with plenty of devices such as smart-home devices connected, the performance gap can be possibly further shrunk.

