# OpenReview forum: "Overcoming Resource Constraints in Federated Learning: Large Models Can Be Trained with only Weak Clients"
_TMLR — Accepted by TMLR_

### Review · Reviewer_phjK · 2023-07-03

**Summary Of Contributions:**

The paper proposes a new method in the federated learning setting where clients have limited computation resources and don't share the intermediate result or labels with the server. Specifically, the method includes first decomposing the model parameter using SVD and then ranking the singular value. Then every client just samples according to the rank based on its computation budget and only updates that main component. The experimental results on several datasets, and domains in both the homogeneous and heterogeneous settings show the proposed method could achieve less performance drop compared with other baselines. It has also included several ablation studies to show the importance of sampling, different sampling strategies, and runtime.

**Audience:**

Yes

**Broader Impact Concerns:**

I don't see any ethical concerns.

**Claims And Evidence:**

Yes

**Requested Changes:**

1. Have some basic theoretical analysis to show the convergence with certain approximation errors.

**Strengths And Weaknesses:**

Pros:
1. The paper is clearly well-written and easy to follow. The background and method have been explained clearly in detail.
2. The paper conducts many experiments in different settings and has some empirical analysis.
3. The experimental results show the proposed method could significantly achieve less performance drop compared with other baselines.

Cons:
1. The proposed method's novelty is not very significant. As mentioned by the related work, using the principle component in federated learning has been used in federated learning several times. The main improvement is using an important sampling method other than the dropout.
2. The proposed method mainly focuses on experimental evidence to show its effectiveness. It lacks a theoretical analysis of the convergence and how well the main principle component could approximate the original weight.

---

### Review · Reviewer_N2vX · 2023-07-04

**Summary Of Contributions:**

The authors focus on the cross-device federated learning setting. They propose to decompose parameters of neural networks into sub-models, this method facilitates the participation of weak clients by reducing their model size. To achieve this, the global model is decomposed into a sum of rank-1 matrices, and each part is randomly assigned to individual clients. Additionally, the authors present numerical illustrations of their algorithm, providing practical insights on its performance.

**Audience:**

Yes

**Broader Impact Concerns:**

I don't see any major concerns, the paper explores a new methodology for federated learning under resource constraints.

**Claims And Evidence:**

Yes

**Requested Changes:**

**Clarifications:**

* In the paragraph below the ``Motivation'' section, there seems to be overlapping definitions between the dimensionality of variable $X$, which is $M \times H \times W$, and the network parameter $W$.

* The step "Sub-model aggregation" and equation (5) are little unclear. I don't understand the definition of the brackets in  $[\text{Sum}(c\in \mathcal{C}) \alpha_i^c u_i^c; \hat{u}_i]$.

* In Algorithm 1, I think the ``orthogonal space refresh'' step is written twice (step 2 and step 10).


**Strengths And Weaknesses:**

**STRENGTHS.**

* The paper is written in a concise style, with clear statements that are easily identifiable in the abstract.

* The idea of decomposing the parameters using SVD (Singular Value Decomposition) appears promising for mitigating the memory and computing bottlenecks faced by edge clients.

* The paper discusses the choice of hyperparameters, including insights on when to recalculate the orthogonal decomposition during training.

**WEAKNESSES.**

* Random assignment of orthogonal vectors may not be suitable to every client. Is it possible to have local parameters that diverge rapidly during training? Unless the parameter matrice is well approximated be few vectors (the spectrum decays rapidly), does this method work with very small sub-models?

---

### Review · Reviewer_o159 · 2023-07-07

**Summary Of Contributions:**

This paper presents PriSM, a novel method aimed at enhancing the efficiency of heterogeneous federated learning algorithms. The central premise of PriSM involves using the singular values of each layer's model weight to guide model sampling, which could range from hidden neurons to convolutional kernels. The methodology was put to the test through numerous federated tasks and models, with the results underscoring the potency of PriSM. In comparison to prevalent benchmark methods that utilize low-rank approximation and structured/ordered dropout, PriSM demonstrated superior performance.

**Audience:**

Yes

**Broader Impact Concerns:**

There is no concern about the ethical implications of the work.

**Claims And Evidence:**

Yes

**Requested Changes:**

- Results of more efforts careful hyperparameter sweeping and optimization on both PriSM and FjORD are expected to be shown in the paper.
- More results on larger-scale language models, e.g., BERT and GPT are expected to be presented.
- Some important works on low-rank model training (though not directly related to FL) shall be discussed.

[1] https://proceedings.mlsys.org/paper_files/paper/2023/hash/b38f833dd45718cc414bd9b2a1d86c9a-Abstract-mlsys2023.html
[2] https://proceedings.mlsys.org/paper_files/paper/2021/hash/94cb28874a503f34b3c4a41bddcea2bd-Abstract.html

**Strengths And Weaknesses:**

Strengths:
- The proposed PriSM method is easy to follow, and the presentation of this paper is concise and clear.
- Adapting FL methods to hardware-heterogeneous environments is of great importance and can potentially attain a lot of attention.
- The final results (especially for the end models' accuracy) are promising.

Weaknesses:
- My major concern is that it's still not clear to me why PriSM can significantly outperform the ordered dropout method, e.g., FjORD even after reading the explanations in Section 4.4. Overall, frankly speaking, PriSM is really similar to FjORD (except for the sampling scheme). So I wonder, after more efforts on careful hyperparameter sweeping and optimization, would the performance gap between PriSM and FjORD become smaller?
- Most of the experimental results are focusing on CNNs running for vision tasks. I wonder if more language experiments can be performed, e.g., BERT/GPT fine-tuning on generative tasks.
- It looks like there is a smarter way to adaptively select r during the entire model training, e.g., gradually using smaller rs during training. So I wonder if the method proposed in automatic low-rank training can be used in PriSM [1].
- I wonder how the proposed PriSM method compare to the depth-wise model splitting approach proposed in [2].

[1] https://proceedings.mlsys.org/paper_files/paper/2023/hash/b38f833dd45718cc414bd9b2a1d86c9a-Abstract-mlsys2023.html
[2] https://arxiv.org/abs/2303.04887

---

### Decision · Action_Editors · 2023-09-25

**Recommendation:** Accept as is

**Comment:**

This work considers decomposing large models in cross-device federated settings into smaller sub-models, which are low-rank approximations of the large model on the server.

All reviewers found the empirical results of the proposed method (PriSM) compelling, particularly when considering highly resource-constrained regimes where each client can only maintain a low capacity model. Reviewers phjK and 0159 mention concerns around a lack of theoretical guarantees and novelty of the approach relative to prior work that similarly explores low-rank training in both federated and non-federated settings.

While I tend to agree with the reviewers that the techniques themselves are somewhat limited in novelty, I also agree with the authors regarding the significance of the proposed techniques: The authors have carefully compared to prior work in their experiments, and the proposed method seems to clearly lead to substantial empirical improvements, particularly in resource-constrained settings. I believe the work is thus worth showcasing at TMLR even in light of the mentioned limitations around novelty and lack of theoretical guarantees.

However, I would encourage the authors to carefully incorporate the discussion from the rebuttal period around differences between this method and prior work (particularly HeteroFL and FjORD) in their revision, as this is helpful for more clearly putting the work in context and understanding the methodological modifications that lead to improved empirical performance.

**Audience:**

Yes, this work focuses developing methodology for resource-constrained federated learning, which is of interest to the TMLR community.

**Claims And Evidence:**

This work develops a method, PriSM, for low-rank training of large models in federated settings. Although reviewers believe the method itself is somewhat limited in novelty, there was also agreement that the empirical improvements of the method relative to existing baselines are compelling.

---

> ### Author Response · Authors · 2023-10-06
> **Camera-ready revision are submitted**
>
> Dear Action Editors,
>
> We sincerely appreciate your time and efforts during the reviewing process, and thank you for your feedback on the work.
>
> We have uploaded the camera-ready revision (with the code released). In the camera-ready revision, we have included a discussion regarding the difference between PriSM and prior works (HeteroFL and FjORD). Please let us know if there is any remaining concerns.
>
> Best, Authors